# *Helichrysum stoechas* (L.) Moench Inflorescence Extract for Tomato Disease Management

**DOI:** 10.3390/molecules28155861

**Published:** 2023-08-03

**Authors:** Eva Sánchez-Hernández, Javier Álvarez-Martínez, Vicente González-García, José Casanova-Gascón, Jesús Martín-Gil, Pablo Martín-Ramos

**Affiliations:** 1Department of Agricultural and Forestry Engineering, ETSIIAA, Universidad de Valladolid, Avenida de Madrid 44, 34004 Palencia, Spain; eva.sanchez.hernandez@uva.es (E.S.-H.); javier.alvarez.martinez@uva.es (J.Á.-M.); jesus.martin.gil@uva.es (J.M.-G.); 2Department of Agricultural, Forest and Environmental Systems, Agrifood Research and Technology Centre of Aragón, Instituto Agroalimentario de Aragón—IA2 (CITA-Universidad de Zaragoza), Avda. Montañana 930, 50059 Zaragoza, Spain; vgonzalezg@aragon.es; 3Instituto Agroalimentario de Aragón—IA2 (CITA-Universidad de Zaragoza), EPS, Universidad de Zaragoza, Carretera de Cuarte s/n, 22071 Huesca, Spain; jcasan@unizar.es

**Keywords:** antifungal activity, biorational, GC–MS, Mediterranean strawflower, natural product, tomato protection

## Abstract

*Helichrysum stoechas* is a singular halophyte that has been shown to have anti-inflammatory, antioxidant, and allelopathic properties. In the work presented herein, we have characterized its inflorescences hydromethanolic extract and assessed its antifungal activity for the pre- and postharvest management of tomato crop diseases. Gas chromatography–mass spectrometry characterization of the extract showed that 4-ethenyl-1,3-benzenediol, 2,3-dihydro-benzofuran, quinic acid, 3,5-dihydroxy-6,7,8-trimethoxy-2-phenyl-4H-1-benzopyran-4-one, 1,6-anhydro-*β*-D-glucopyranose, catechol, scopoletin, and maltol were the main constituents. The co-occurrence of pyranones, benzenediols, and quinic acids as phytoconstituents of *H. stoechas* extract resulted in promising in vitro minimum inhibitory concentrations of 500, 375, 500, 187.5, 187.5, and 375 μg·mL^−1^ against mycelia of *Alternaria alternata*, *Colletotrichum coccodes, Fusarium oxysporum* f. sp. *lycopersici, Rhizoctonia solani, Sclerotinia sclerotiorum*, and *Verticillium dahliae*, respectively. Further, to assess the potential of *H. stoechas* inflorescence extract for postharvest tomato crop protection, ex situ tests were conducted against *C. coccodes*, obtaining high protection at a dose of 750 μg·mL^−1^. Taking into consideration that the demonstrated activity is among the highest reported to date for plant extracts and comparable to that of the synthetic fungicides tested as positive controls, *H. stoechas* inflorescence extract may be put forward as a promising biorational and may deserve further testing in field-scale studies.

## 1. Introduction

The genus *Helichrysum* comprises up to 600 species of flowering plants in the *Asteraceae* family. *Helichrysum* spp. have been utilized in various folk medicinal systems for addressing fever and inflammation and managing neurologic and digestive disorders [1,2]. Certain healing attributes have been validated by medical science, including its antimicrobial activity [3,4].

In particular, *Helichrysum stoechas* (L.) Moench, known as Mediterranean strawflower, curry plant, or yellow amaranth, is a fragrant, thermophilous halophyte found in southern Europe. It is a perennial or annual shrub that likes dry, rocky, and sandy areas. It is a hermaphrodite with grayish-green foliage and yields petite spherical yellow inflorescences. 

Phytochemical studies of *Helichrysum* plants have revealed their richness in phenolic compounds (flavonoids, phloroglucinols, and pyrones), and some species also contain terpenes [5,6]. For instance, the hydroalcoholic extract of *H. stoechas* is rich in 3,5-dicaffeoylquinic acid, myricetin and quercetin glucosides, and acetylhexosides [7]. 

*Helichrysum stoechas* has demonstrated anti-acetylcholinesterase, anti-tyrosinase, anti-α-glucosidase, and antioxidative properties [8]. As for its antimicrobial activity, its ethanol extract, which contains caffeoylquinic acid and dicaffeoylquinic acid isomers, together with kaempferol, quercetin, and naringerin glucosides, has antimicrobial activity against *Enterobacter cloacae*, *Escherichia coli*, *Klebsiella pneumoniae*, *Pseudomonas aeruginosa*, and *Staphylococcus aureus* [9]. Fractionation of its dichloromethane extract yielded *β*-sitosterol-*β*-*O*-glucosides, 4-hydroxy-3(isopentel-2-yl) acetophenone, italipyrone, plicatipyrone, and helipyrone, with an antimicrobial effect on Gram-positive bacteria [10]. Likewise, the essential oils obtained from *H. stoechas*—rich in *α*-humulene, *α*-pinene, *β*-caryophyllene, and limonene—showed activity against Gram-positive bacteria (*S. aureus* and *Staphylococcus epidermis*), Gram-negative bacteria (*E. coli, E. cloacae, K. pneumoniae,* and *P. aeruginosa*), and pathogenic yeasts (*Candida albicans* (C.P.Robin) Berkhout, *Candida tropicalis* Berkhout, and *Nakaseomyces glabratus* (H. W. Anderson) Sugita & Takashima [11]). 

The aforementioned antimicrobial activity against human pathogens makes *H. stoechas* a promising candidate for valorization for crop protection, offering natural and eco-friendly alternatives to synthetic pesticides. For instance, *H. stoechas* could be used for tomato (*Solanum lycopersicum* L.) protection against bacterial and fungal diseases and serve as an organic substitution for artificial preservatives due to its antioxidant properties [8]. 

The feasibility of a natural biorational-based approach has previously been demonstrated against early blight disease caused by *Alternaria solani* Sorauer. Instead of costly and hazardous chemical fungicides that pose health and environmental risks [12] and may lead to the development of fungicide-resistant strains [13], phenolic-rich plant extracts have effectively combated various *Alternaria* species [14,15,16,17].

Building upon this knowledge, in this study, we aimed to investigate the antifungal properties of *H. stoechas* hydromethanolic extract against six important tomato fungal pathogens. Apart from *A. alternata*, the in vitro activity was also tested for the control of root and foot rot (caused by *Rhizoctonia solani* Kuhn) [18], sclerotinia stem rot (caused by *Sclerotinia sclerotiorum* (Lib.) De Bary) [19], Verticillium wilt (caused by *Verticillium dahliae* Kleb.) [20], and Fusarium wilt (caused by *Fusarium oxysporum* f. sp. *lycopersici* (Sacc.) Snyder & Hansen) [21]. Further, to assess the potential of *H. stoechas* extract for postharvest tomato crop protection, in vitro and ex situ tests were also conducted against *Colletotrichum coccodes* (Wallr.) Hughes, which causes the black dot or anthracnose rot [22]. The presented results contribute to the development of sustainable control strategies in horticulture, addressing the need for effective alternatives to synthetic fungicides while ensuring food security and crop health.

## 2. Results

### 2.1. Infrared Spectroscopy Characterization

The main bands of the infrared spectrum of *H. stoechas* dried inflorescence samples (Appendix A) and their assignments are summarized in Table 1. The functional groups found are in line with the chemical constituents detected using gas chromatography–mass spectrometry, GC–MS (explained later). Specifically, the spectrum contains absorption bands also seen in the infrared spectra of those phytochemicals. For example, the absorption band at 984 cm^−1^ (vinyl groups vibration) in 4-ethenyl-1,3-benzenediol; those at 1178 cm^−1^ and 596 cm^−1^ in 2,3-dihydro-benzofuran; the one at 1688 cm^−1^ in quinic acid; those at 1652 cm^−1^, 1444 cm^−1^, 1367 cm^−1^, and 1263 cm^−1^ in 3,5-dihydroxy-6,7,8-trimethoxy-2-phenyl-4*H*−1-benzopyran-4-one; that at 1116 cm^−1^ in 1,6-anhydro-*β*-d-glucopyranose; or those at 1514 cm^−1^ and 853 cm^−1^ in scopoletin have been observed. The band at 1597 cm^−1^ is shared by 2,3-dihydro-benzofuran, 3,5-dihydroxy-6,7,8-trimethoxy-2-phenyl-4*H*−1-benzopyran-4-one, and 1,6-anhydro-*β*-d-glucopyranose.

### 2.2. GC–MS Characterization

GC–MS chromatogram of the *H. stoechas* inflorescence extract (Appendix A) includes the phytochemicals presented in Table 2. As shown in Figure 1, the main chemical species were 4-ethenyl-1,3-benzenediol (10.4%); 2,3-dihydro-benzofuran (5.8%); quinic acid (5.6%); 3,5-dihydroxy-6,7,8-trimethoxy-2-phenyl-4*H*-1-benzopyran-4-one (5.1%); 1,6-anhydro-*β*-d-glucopyranose (4.6%); catechol (3.5%); scopoletin (2.9%); 4-pyrimidinol, 6-(methoxymethyl)-2-(1-methylethyl)- (2.6%); 2-hydroxy-*γ*-butyrolactone (2.6%); 6-methyl-3(2*H*)-pyridazinone (2.4%); maltol (2.4%); 1-acetyl-2-amino-3-cyano-7-isopropyl-4-methylazulene (2.2%); *α*-bisabolol (or levomenol, 2.1%); 2,3-dihydro-3,5-dihydroxyl-6-methyl-4*H*-pyran-4-one (2%); and octadec-9-enoic acid (2%). 

### 2.3. Antifungal Activity

#### 2.3.1. In Vitro Antifungal Activity

The antifungal susceptibility test results are depicted in Figure 2. In all instances, an increase in *H. stoechas* extract concentration resulted in a decrease in mycelium radial growth, yielding statistically significant variances. *R. solani* and *S. sclerotiorum*, specifically, exhibited the highest sensitivity to *H. stoechas* inflorescence hydromethanolic extract, with minimal inhibitory concentrations (MICs) of 187.5 μg·mL^−1^. Complete inhibition for *C. coccodes* and *V. dahliae* mycelial growth occurred at 375 μg·mL^−1^, while a higher dosage of 500 μg·mL^−1^ was required to inhibit *A. alternata* and *F. oxysporum* f. sp. *lycopersici* growth. Table 3 displays the effective concentrations at 50% and 90% (EC_50_ and EC_90_, respectively).

Results of mycelial growth inhibition for the three commercial fungicides selected as positive controls are summarized in Table 4. The mancozeb dithiocarbamate fungicide, at a dosage of 150 μg·mL^−1^ (one-tenth of the recommended amount), exhibited the highest efficacy, inhibiting the growth of all plant pathogens except for *A. alternata*. At the recommended concentration of 2000 μg·mL^−1^, fosetyl-Al organophosphorus fungicide completely inhibited the growth of all fungal species except for *A. alternata, F. oxysporum* f. sp. *lycopersici*, and *S. sclerotiorum*. Conversely, the strobilurin fungicide (azoxystrobin), at a recommended dose of 62,500 μg·mL^−1^, displayed the lowest efficacy, failing to fully arrest the development of all phytopathogens.

#### 2.3.2. Ex Situ Postharvest Protection Tests

*H. stoechas* inflorescence extract was assessed as a protective measure against anthracnose on tomato cv. “Daniela” fruits. Two concentrations were tested: MIC and MIC×2 (375 and 750 μg·mL^−1^, respectively). The results are displayed in Figure 3 and Figure 4. In the positive control (*C. coccodes* inoculated on tomato fruits and treated solely with bidistilled water), fruits showed dark brown, circular, sunken lesions around the inoculation zone, delimited by a circular chlorotic halo and displaying evident soft rot symptoms ten days post-inoculation (Figure 4b). The average lesion diameter was 42.2 ± 3.7 mm (Table 5). *H. stoechas* inflorescence extract, at the MIC concentration, inhibited anthracnose on the fruit by 27%, resulting in lesions similar to the positive control (Figure 4c). However, when the extract was applied at a higher concentration (MIC×2, Figure 4d), anthracnose symptoms were inhibited by >80% compared to the positive control. 

## 3. Discussion

### 3.1. On the Phytochemical Profile

Considering the hydromethanolic extraction mixture’s ability to solubilize non-volatile polar compounds that cannot be detected without previous derivatization before carrying out the GC–MS analysis, it is important to exercise caution with the results. In this study, such prior derivatization was not conducted due to drawbacks such as increased procedural preparation time and cost (which would have a negative impact on the economic viability of the crop protection treatment), complex data acquisition, potential impurities, uncertain compound conversion into derivatives, and the use of toxic reagents [25]. On the other hand, the injection of non-volatile compounds may result in eventual damage to the capillary column.

Regarding the reliability of GC–MS identification of extract components, limitations in identifying certain minority compounds were observed, with low quality of resemblance (Qual) values. This suggests that the identification of compounds like 4-pyrimidinol, 6-(methoxymethyl)-2-(1-methylethyl)-; 2-hydroxy-*γ*-butyrolactone; and 1-acetyl-2-amino-3-cyano-7-isopropyl-4-methylazulene may hold some value, but accuracy cannot be guaranteed. The main constituents, except for 4-ethenyl-1,3-benzenediol, had Qual values higher than 87. In the case of this chemical species, identified at a retention time (RT) of 12.7793 min and for which a Qual = 64 was obtained using the NIST11 database, reintegration and indexing using the Wiley database confirmed its presence (Appendix A shows a good MS agreement), also supported by infrared vibrational data.

As for the prior findings on the identified phytochemicals, 4-ethenyl-1,3-benzenediol (or 4-vinylresorcinol) is connected to resveratrol (5-[(1E)-2-(4-hydroxyphenyl)ethenyl]-1,3-benzenediol) and is a stress metabolite (phytoalexin) produced by *Vitis vinifera* L. [26,27]. 2,3-Dihydrobenzofuran is found in *Phyla nodiflora* (L.) Greene, *V. vinifera,* and *Citrullus colocynthis* (L.) Schrader [28] and is widely distributed in higher plants, mainly from the *Asteraceae* family [29]. Quinic acid, related to 3,5-dicaffeoylquinic acid, is a cyclic polyol found in cinchona bark and in plants such as *Gamblea innovans* (Siebold & Zucc.) C.B.Shang, Lowry & Frodin, *Pterocaulon virgatum* (L.) DC. [30], and *Euphorbia serrata* L. [31]. 3,5-Dihydroxy-6,7,8-trimethoxy-2-phenyl-4H-1-benzopyran-4-one is a flavone present in *Helichrysum arenarium* (L.) Moench and *Artemisia klotzchiana* Besser. As regards 1,6-anhydro-*β*-d-glucopyranose (levoglucosan), it is an anhydrohexose found in *Lotus creticus* L., *Lotus filicaulis* Durieu, *Equisetum arvense* L. [32], and *Sambucus nigra* L. [33]. It is employed for making biochemically significant substances like (+)-biotin, indanomycin, macrolide antibiotics, quinone, rifamycin S, tetrodotoxin, and thromboxane B2 [34]. Catechol is a benzenediol whose chemical structure is close to that of 4-ethenyl-1,3-benzenediol. It was detected in *S. nigra* flower extract [33]. Scopoletin is a naturally occurring coumarin derivative (i.e., a 1,2-benzopyrone) found in the roots of *Scopolia* and *Urtica* genera, in flowers of *Passiflora* spp., and in several *Asteraceae*. Maltol is a hydroxypyranone that can be located in pine needles and larch tree bark.

Concerning the chemical profile of the *H. stoechas* inflorescences extract, important phytochemicals have been pyranones, such as 3,5-dihydroxy-6,7,8-trimethoxy-2-phenyl-4H-1-benzopyran-4-one, scopoletin, and maltol, and phenolic acid derivatives such as quinic acid. These components do not coincide exactly with those identified by Barroso et al. [7] (quercetin/myricetin and caffeoylquinic acid), but they have an obvious structural analogy. Phytoconstituents not evidenced in previous reports on *H. stoechas* extracts have been 4-ethenyl-1,3-benzenediol, 2,3-dihydro-benzofuran, 1,6-anhydro-*β*-D-glucopyranose, and catechol, all with potential antimicrobial properties [35,36,37]. These differences may be tentatively attributed either to variations in the extraction procedure or to individual, genotype-depending differences, location-related intra-varietal differences, and seasonal variations—all of which could significantly influence phytochemical composition and bioactivity. Additionally, the existence of different chemotypes due to minor genetic and epigenetic changes cannot be excluded. In this regard, analyzing the stability and repeatability of the occurrence of individual components would be an essential area of investigation. This subject has not been covered in the study presented herein or in other previous studies on *H. stoechas* [7,8,9,10,11], highlighting its potential as a line for further research. 

As regards bactericide and fungicide activities of other phytochemicals identified in the *H. stoechas* inflorescence extract, there are references on the activities of scopoletin [38,39], maltol [40], and quinic acid [41,42]. Scopoletin inhibits Gram-positive bacteria, such as *Enterococcus faecium* and *S. aureus* (MIC = 128 μg mL^−1^), as well as Gram-negative bacteria, such as *Stenotrophomonas maltophila* (MIC = 256 μg·mL^−1^), and quinic acid derivatives were effective against fungi [43] and *S. aureus* [44].

### 3.2. Antifungal Activity

#### 3.2.1. Comparison with Other Helichrysum spp. Extracts

*H. stoechas* has been tested for antifungal activity against human pathogens, exhibiting MIC values of 8 µg·mL^−1^ against *C. albicans* and *Candida parapsilosis* (Ashford) Langeron & Talice for aqueous and ethanol extracts from its aerial parts [4]. Sobhy et al. [9] reported that the *H. stoechas* apical parts essential oil (0.7% *v*/*w*) inhibited *C. albicans*, but not *C. tropicalis* and *N. glabratus* (the ethanolic extract showed no inhibitory activity). However, Roussis et al. [11] discovered that the essential oil derived from the aerial organs of *H. stoechas* was effective against *C. albicans, C. tropicalis*, and *N. glabratus*, with MIC values in the 3.25–6.8 µg·mL^−1^ range.

In contrast, the antimicrobial activity of related species such as *Helichrysum odoratissimum* (L.) Sw., *Helichrysum patulum* (L.) D.Don, *Helichrysum italicum* (Roth) G. Don, and *Helichrysum plicatum* DC has been tested against phytopathogenic fungal taxa. Matrose et al. [45] examined the antifungal efficacy of *H. odoratissimum* and *H. patulum* ethanol extracts against *Botrytis cinerea* Pers., observing inhibition percentages of 65% and 51%, respectively, at a dosage of 250,000 μg·mL^−1^. The essential oil from the aerial parts of *H. italicum* was tested against four fungi (namely, *A. alternata*, *Ascochyta rabiei* (Pass.) Labr., *Aspergillus niger* Tiegh., and *Fusarium solani* var. *coeruleum* (Lib. ex Sacc.) C.Booth) [46], finding fungistatic MICs in the 6.325 to 50.6 μg·mL^−1^ range (lower than that reported here). Regarding the antimicrobial properties of the aqueous ethanol extract of *H. plicatum*, it inhibited the growth of most tested fungi (including *A. alternata*, *Aspergillus flavus* Link, *Chaetomium* sp., *Curvularia lunata* (Wakker) Boedijn, *Fusarium equiseti* (Corda) Sacc., *Fusarium solani* (Mart.) Sacc., *Fusarium subglutinans* (Wollenw. & Reinking) P.E. Nelson, Toussoun & Marasas, *Fusarium verticillioides* (Sacc.) Nirenberg, and *Penicillium* spp.) at concentrations in the 5–40 μg·mL^−1^ range [47], which are also lower than the MIC reported for *A. alternata* in this study. 

While both hydromethanolic plant extracts and essential oils can be effective at controlling phytopathogens, it is worth noting that hydroalcoholic plant extracts (such as the one discussed in this work) have some advantages over essential oils, including a broader spectrum of activity, less phytotoxicity, easier extraction, and greater stability.

#### 3.2.2. Comparison of Efficacy vs. Other Plant Extracts

A more extensive comparison with the effectiveness of other tested plant extracts against the six fungi studied herein can be found in Appendix A [23,24,33,48,49,50,51,52,53,54,55,56,57,58,59,60,61,62,63,64,65,66,67,68,69,70,71,72,73,74,75,76,77,78,79,80,81,82,83,84,85,86,87,88,89,90,91,92,93,94,95,96,97,98,99]. However, caution should be exercised in comparing the results due to variations in isolates (or species, in the case of the genus *Colletotrichum*) across different studies. Further, in studies where multiple plant extracts were tested, those lacking activity were excluded.

Regarding *A. alternata*, the *H. stoechas* extract demonstrated the highest activity (MIC = 500 µg·mL^–1^) among the reported literature, except for the aqueous ethanol extract of *H. plicatum* mentioned above. As for the activity against *C. coccodes* (MIC = 375 µg·mL^−1^), no direct comparisons were available, but the activity would be among the highest against *Colletrotrichum* spp., together with those of *Zingiber officinale* Roscoe rhizomes chloroform extract and *Polyalthia longifolia* (Sonn.) Thwaites leaves methanol extract, for which inhibition rates of 87.4 and 84% were attained at 400 µg·mL^−1^ [61]. 

Regarding *F. oxysporum* (MIC = 500 µg·mL^−1^), its effectiveness was comparable to the ethyl acetate and methanol extracts of *Cestrum nocturnum* L. flowers (MIC = 500 µg·mL^–1^) [51]. Against *V. dahliae*, *H. stoechas* extract demonstrated the highest activity (MIC = 375 µg·mL^−1^), followed by an *Uncaria tomentosa* (Willd. ex Schult.) DC. aqueous ammonia bark extract (500 µg·mL^−1^) [24]. Concerning *R. solani*, the activity of *H. stoechas* was the highest (MIC = 187.5 µg·mL^−1^), followed by the chloroform extracts of *Clerodendrum infortunatum* L. leaves and *Z. officinale* rhizomes, as well as the methanol extract of *P. longifolia* leaves, all of which achieved complete inhibition at 400 µg·mL^−1^ [61]. Concerning *S. sclerotiorum*, the *H. stoechas* inflorescence extract was the second most effective (MIC = 187.5 µg·mL^−1^) after the crude aqueous extract of *Rhus coriaria* L. fruit [100] (MIC = 5 µg·mL^−1^). Notably, its inhibitory activity surpassed that of the ethyl acetate extract of *C. nocturnum* flowers (MIC = 250 µg·mL^–1^) [51].

#### 3.2.3. Conventional Fungicide Comparison

When the antifungal activity of *H. stoechas* inflorescence extract (Table 3) was compared with that of conventional synthetic fungicides (Table 4), it was found that the extract was generally less effective than mancozeb against all pathogens, except for *A. alternata*. In the case of this pathogen, *H. stoechas* extract achieved full inhibition at 500 µg·mL^–1^, whereas mancozeb required over 1500 µg·mL^–1^. *H. stoechas* extract led to complete inhibition at concentrations lower than the recommended dose of fosetyl-Al (2000 µg·mL^–1^). Nevertheless, fosetyl-Al was more effective against *C. coccodes*, with complete inhibition observed at 200 µg·mL^–1^ vs. 375 µg·mL^–1^ for *H. stoechas* extract. Fosetyl-Al, even at the recommended dose, did not fully inhibit *A. alternata, F. oxysporum* f. sp. *lycopersici,* and *S. sclerotiorum*, requiring doses higher than 2000 µg·mL^–1^. At the prescribed concentration of 62,500 µg·mL^–1^, azoxystrobin failed to completely hinder any of the six fungal pathogens, indicating notably lower efficacy than the plant extract.

#### 3.2.4. Postharvest Protection Tests

Hydromethanolic plant extracts have not been tested for ex situ inhibition of tomato anthracnose caused by *C. coccodes* or other *Colletotrichum* spp. Regarding alternative extraction media, *R. coriaria* aqueous crude extract at 20 µL·mL^−1^ provided complete protection of tomato fruits against *Colletotrichum acutatum* J.H.Simmonds after ten days of incubation [100], indicating higher efficacy compared to *H. stoechas* extract. 

In studies involving other fruits, the aerial parts extract of *Cymbopogon winterianus* Jowitt ex Bor (at 1500 µL·mL^−1^, twice the dosage tested in this research) significantly outperformed mancozeb (2500 µL·mL^−1^) in controlling the artificial infection of banana (*Musa* × *paradisiaca*) fruits with *Colletotrichum musae* (Berk. & M.A. Curtis) Arx. [101]. Additionally, Necha et al. [77] examined twelve plant extracts (at a dosage of 2:10 *w/v*) for protecting *Carica papaya* L. and *Mangifera indica* L. fruits against *Colletotrichum gloeosporioides* (Penz.) Penz. & Sacc. The papaya leaf extract provided full protection for papaya fruits, while *Pouteria sapota* (Jacq.) H.E. Moore and Stearn leaf extract resulted in 20% infection. In mango fruits, the stem extracts of *Annona reticulata* L., *Dyospiros ebenaster* Retz, and *Tamarindus indicus* L. offered the highest level of protection, with only 10% infection. 

Concerning essential oils, cinnamon and clove ones were found to reduce lesion diameter on immature green pepper fruits inoculated with *C. gloeosporioides* [102]. In another study [103], cinnamon and lemongrass oils were reported to exhibit strong inhibitory activity against *C. acutatum* on mangoes but caused severe damage to fruit peels, while basil essential oil reduced *C. acutatum* lesions without harming the fruit.

In terms of innovative application methods, nanoemulsion-based coatings have been proposed as an effective technology for anthracnose control [104]. For example, Oliveira et al. [105] demonstrated that coatings combining chitosan (at 5000 µg·mL^−1^) with *Cymbopogon citratus* (D.C. ex Nees) Stapf essential oil (0.15–0.6 µL·mL^−1^) exhibited similar or even better efficacy than synthetic fungicides in controlling anthracnose on guava (*Psidium guajava* L.), mango, and papaya 12 days after inoculation. Similarly, Grande Tovar et al. [106] and Peralta-Ruiz et al. [107] investigated the inhibitory effects of chitosan and *Ruta graveolens* L. essential oil coatings on guava and papaya fruits infected with *Colletotrichum* spp., and observed reductions in lesion expansion ranging from 50–67% for a treatment dose of 0.5% to 69–100% for concentrations of 1–1.5%. 

These promising findings suggest that *H. stoechas* extract could be incorporated into chitosan films and coatings in future studies, benefiting from potential synergistic interactions with the biopolymer. Such films and coatings, applied via spray coating or fruit dipping, would be more reproducible and scalable treatment methods for potential industrial application than the one assayed herein. In this regard, regardless of whether the extract is used alone or dispersed in biopolymeric films, the development of formulations based on *H. stoechas* extracts would require further research and exploration at a more advanced stage.

## 4. Material and Methods

### 4.1. Reagents and Fungal Isolates

Potato dextrose broth (PDB) and potato dextrose agar (PDA) came from Becton, Dickinson, and Company (Franklin Lakes, NJ, USA). Tween^®^ 20 (CAS No. 9005-64-5) was bought from Sigma Aldrich Quimica S.A. (Madrid, Spain).

To conduct the in vitro experiments, we used certain fungicides as positive controls. These included Ortiva^®^ (azoxystrobin 25%; Syngenta, Basel, Switzerland), Vondozeb^®^ (mancozeb 75%; UPL Iberia, Barcelona, Spain), and Fesil^®^ (fosetyl-Al 80%; Bayer, Leverkusen, Germany), kindly provided by the Plant Health and Certification Center (CSCV) of the Gobierno de Aragón.

The fungal isolates of *A. alternata* (CRD 41/37/2019), *C. coccodes* (CRD 246/190), and *R. solani* (CRD 207/99) were obtained from the Regional Diagnostic Center of Aldearrubia (Junta de Castilla y León). *S. sclerotiorum* (MYC-799) and *V. dahliae* (MYC-1134) were acquired from the Centre for Agrifood Research and Technology of Aragon (CITA). Additionally, *F. oxysporum* f. sp. *lycopersici* (CECT 2866) was obtained from the Spanish Type Culture Collection (Valencia, Spain).

### 4.2. Plan Material and Extraction Protocol

Aerial parts were collected from *H. stoechas* plants in June 2022 near the city of Huesca, Spain. The specific location was 42°09′15.4″ N 0°27′50.1″ W. The plants were in full bloom at that time. A voucher specimen, verified by Prof. J. Ascaso, was stored in the herbarium of EPS–Universidad de Zaragoza. The inflorescences were separated from stems and leaves. To create representative composite samples, 20 specimens were mixed together. These composite samples were dried in the shade, ground into a fine powder using a mechanical grinder, and then homogenized and sieved through a 1 mm mesh.

The extraction process using ultrasonication was similar to the one described in [31]. The use of a methanol:water (1:1, *v*/*v*) extraction medium offers versatility, cost-effectiveness, and efficient extraction of a wide range of phytochemicals. Ultrasound-assisted extraction provides increased extraction efficiency, reduced extraction time, preservation of compound integrity, and energy efficiency. The procedure was as follows: the dried inflorescence sample (19.6 g) was mixed with a methanol:water solution (1:1 *v*/*v*; 250 mL). The mixture was heated and stirred for 20 min at 50 °C. It was then sonicated using a model UIP1000 hdT probe-type ultrasonicator from Hielscher Ultrasonics (Teltow, Germany). After sonication, the mixture was centrifuged at 9000 rpm for 10 min. The resulting liquid was filtered through Whatman No. 1 paper and freeze-dried, resulting in a solid residue. The extraction yield was only 0.6%.

For the subsequent GC–MS analysis, the freeze-dried extract was redissolved in methanol (HPLC-grade) to yield a solution with a concentration of 5 mg·mL^−1^. The solution was then filtered again.

### 4.3. Characterization Procedures

The infrared vibrational spectrum of the dried inflorescence sample from *H. stoechas* was measured using an iS50 Fourier-transform infrared (FTIR) spectrometer (Nicolet, Thermo Scientific; Waltham, MA, USA) with an attenuated total reflectance (ATR) system. The range of measurement was 400–4000 cm^−1^, with a 1 cm^−1^ resolution. The resulting spectrum was obtained by combining 64 scans.

The hydroethanolic extract of *H. stoechas* inflorescence was analyzed using a GC–MS system at the Research Support Services of Universidad de Alicante. The system consisted of a 7890A gas chromatograph coupled to a 5975C quadrupole mass spectrometer (Agilent Technologies; Santa Clara, CA, USA). The following conditions were used for chromatography: injection volume = 1 µL; injector temperature = 280 °C (in splitless mode); and initial oven temperature = 60 °C for 2 min, followed by a ramp of 10 °C/min up to a final temperature of 300 °C for 15 min. Separation of compounds was achieved using an HP-5MS UI column (Agilent Technologies) with a length of 30 m, a diameter of 0.250 mm, and a film thickness of 0.25 µm. The mass spectrometer conditions were as follows: temperature of the electron impact source = 230 °C; temperature of the quadrupole = 150 °C; and ionization energy = 70 eV. Components were identified by comparing their mass spectra and retention time with those of authentic compounds and by utilizing the databases of the National Institute of Standards and Technology (NIST11) and Wiley.

### 4.4. In Vitro Antifungal Activity

The antifungal activity of the *H. stoechas* aerial part extract was assessed using the poisoned food method [108]. Stock solution aliquots were added to the PDA medium, resulting in final concentrations ranging from 15.62 to 1500 µg·mL^−1^. Mycelial plugs coming from one-week-old PDA cultures of *A. alternata, C. coccodes, F. oxysporum* f. sp. *lycopersici, R. solani, S. sclerotiorum*, and *V. dahliae* were transferred to plates containing the amended media. Each treatment and concentration combination utilized three plates, with the experiment repeated twice. The untreated control involved replacing the extract with the solvent used for extraction in the PDA medium (methanol:water, 1:1 *v*/*v*). Additional controls including pure PDA medium and PDA with the lowest treatment concentration were included to validate the absence of contamination. Positive controls consisted of commercial fungicides, namely, Ortiva^®^, Vondozeb^®^, and Fesil^®^, and were conducted according to the indications and doses recommended by each manufacturer. 

It was decide to segregate the analysis of fungicides from that of the extract evaluation for several reasons: on the one hand, the recommended concentrations of the commercial products are usually significantly different from those used in laboratory standards for antibiotic activity; on the other hand, commercial fungicide products are typically formulated with specific purity levels and often contain additional substances that enhance their effectiveness (and, consequently, their dose–response curve), while plant extracts are complex matrices of several active components, where the adjustment of the final concentrations employed are made on the whole in each specific extract.

In all bioassays, radial mycelium growth was evaluated by measuring the average of two colony diameters that were perpendicular to each other for every repetition. Growth suppression was determined using the following formula after a one-week incubation in complete darkness at a temperature of 25 °C: ((d_c_ − d_t_)/d_c_) × 100, where d_c_ denotes the mean colony diameter in the untreated control, and d_t_ represents the mean diameter of the treated colony. The effective concentrations were estimated by fitting them to a four-parameter logistic equation (dose–response curve). The mycelial growth inhibition results were analyzed in IBM SPSS Statistics v.25 (IBM; Armonk, NY, USA) using analysis of variance (ANOVA), followed by Tukey’s test for post hoc comparison of means, as the Shapiro–Wilk and Levene tests confirmed homogeneity and homoscedasticity.

### 4.5. Preparation of Conidial Suspension of C. coccodes

A conidial suspension of *C. coccodes* was prepared as per Sánchez-Hernández et al. [109], with minor modifications. Conidia were obtained from 1-week-old PDB cultures (200 mL broth kept in the dark at 25 °C and 140 rpm in an orbital stirrer incubator). The suspension was filtered through two layers of sterile muslin to remove somatic mycelia. Spore concentration was determined using a hemocytometer (Weber Scientific International Ltd.; Teddington, Middlesex, UK), and adjusted to a final concentration of 1 × 10^6^ spores (conidia)·mL^−1^.

### 4.6. Ex Situ Protection of Tomato Fruits

The efficacy of *H. stoechas* extract was assessed on artificially infected tomato fruits (cv. “Daniela”), cultivated according to EU organic farming regulations by Huerta El Gurullo (Cuevas del Almanzora, Almería, Spain). All the assayed fruits had a similar size (about 75 mm in diameter) and showed no visible disease symptoms. We slightly modified the protocol proposed by Wang et al. [110]. First, the tomatoes were surface disinfected for 2 min using a 3% NaOCl solution. Then, they were rinsed three times with sterile distilled water and dried on sterile absorbent paper in a laminar flow hood. The fruits were divided into four groups: one group was treated with *H. stoechas* extract at a concentration equal to the MIC determined in vitro (375 μg·mL^−1^) and another group received twice the MIC concentration (750 μg·mL^−1^), while the remaining two groups served as negative and positive controls (no treatment/no pathogen and pathogen/no treatment, respectively). Under aseptic conditions, each fruit was punctured at three equidistant points in the equatorial region using a truncated needle (3 mm diameter × 5 mm depth). The treated fruits were initially filled with 20 µL of the corresponding treatment (at MIC or MIC×2 concentrations, supplemented with 0.2% Tween^®^ 20). After one hour, wounds were inoculated with 20 µL of a *C. coccodes* spore suspension (1 × 10^6^ conidia·mL^−1^). Positive controls were solely inoculated with the *C. coccodes* spore suspension, while negative controls were inoculated with sterile deionized water containing 0.2% Tween^®^ 20. Each fruit was placed in a separate clean container (corresponding to its treatment) with sterile moistened cotton and incubated at 25 °C for ten days. Lesion diameters were measured twice at right angles to one another on the fruit surfaces, and the percentage of lesion size reduction compared to the positive control (0% reduction) was calculated using the formula: LSR (%) = [(LS_c_ − LS_t_)/LS_c_] × 100, where LS_c_ represents the lesion diameter of the positive control, and LS_t_ represents the lesion diameter of the treated fruits. On day 10, at the end of the experiment, the tomatoes were cut open to analyze the internal lesions.

In these experiments, a contrast fungicide was not used, given that, in the Spanish national legislation on registration of phytosanitary products, there is currently no authorized fungicide for direct use in this plant product (postharvest tomatoes).

## 5. Conclusions

This research examined *Helichrysum stoechas* inflorescence hydromethanolic extract’s antifungal properties as a biocontrol agent for tomato phytopathogens. GC–MS analysis identified various compounds including pyranones, benzenediols, and quinic acids, with 4-ethenyl-1,3-benzenediol, 2,3-dihydro-benzofuran, quinic acid, 3,5-dihydroxy-6,7,8-trimethoxy-2-phenyl-4*H*-1-benzopyran-4-one, 1,6-anhydro-*β*-d-glucopyranose, catechol, scopoletin, and maltol as the key constituents. In vitro tests demonstrated significant activity against *A. alternata, C. coccodes, F. oxysporum* f. sp. *lycopersici, R. solani, S. sclerotiorum*, and *V. dahliae*, with MIC values ranging from 187.5 to 500 μg·mL^−1^, indicating broad-spectrum antifungal behavior. Remarkably, *H. stoechas* extract showed higher activity against *A. alternata* than mancozeb, as well as superior efficacy compared to fosetyl-Al (except against *C. coccodes*) and azoxystrobin. Furthermore, it exhibited one of the most potent antifungal effects among those reported for plant extracts. Notably, as a postharvest treatment for anthracnose, a dose of 750 μg·mL^−1^ of *H. stoechas* extract provided significant protection. These findings underscore the potential of this halophyte as a natural alternative to synthetic fungicides for managing tomato crop fungal diseases.

## Figures and Tables

**Figure 1 molecules-28-05861-f001:**
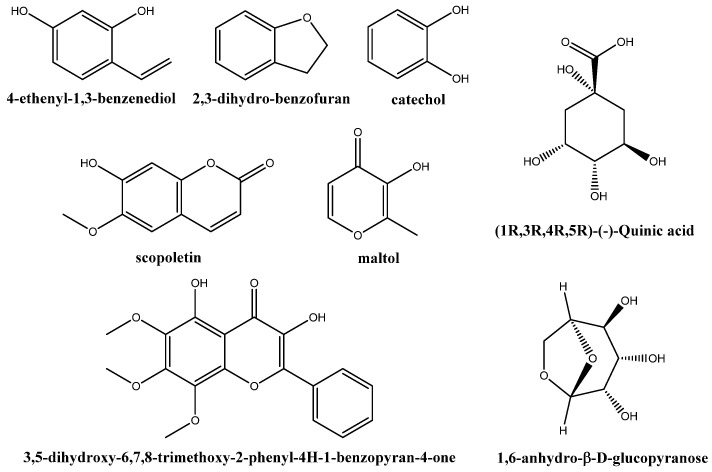
Chemical structures of the main phytochemical compounds identified in *H. stoechas* inflorescence hydromethanolic extract using GC–MS.

**Figure 2 molecules-28-05861-f002:**
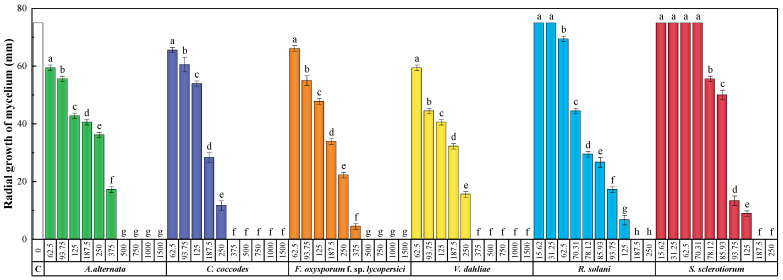
Mycelial growth inhibition achieved with the hydromethanolic extract of *H. stoechas* inflorescences against *A. alternata*, *C. coccodes*, *F. oxysporum* f. sp. *lycopersici*, and *V. dahliae* at concentrations in the 62.5 to 1500 μg·mL^−1^ range (or ranging from 15.62 to 250 μg·mL^−1^ for *R. solani* and *S. sclerotiorum*). Same letters denote non-significant differences at *p* < 0.05. Error bars show standard deviations (*n* = 6). ‘C’ represents the untreated control (each fungus growing in potato dextrose agar, PDA, medium with only the extraction solvent added).

**Figure 3 molecules-28-05861-f003:**
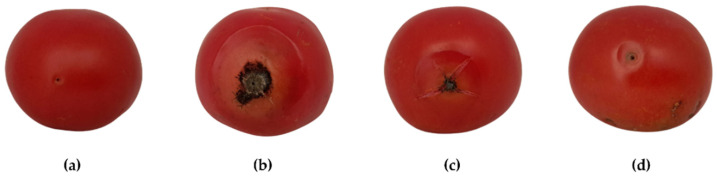
External lesions caused by *C. coccodes* on tomatoes cv. “Daniela” ten days after artificial inoculation in the presence/absence of *H. stoechas* inflorescence extract: (**a**) negative control; (**b**) fruits artificially inoculated with *C. coccodes* (positive control); (**c**) fruits treated with *H. stoechas* extract at 375 µg·mL^−1^; (**d**) fruits treated with *H. stoechas* extract at 750 µg·mL^−1^. Only one replicate per treatment is shown.

**Figure 4 molecules-28-05861-f004:**
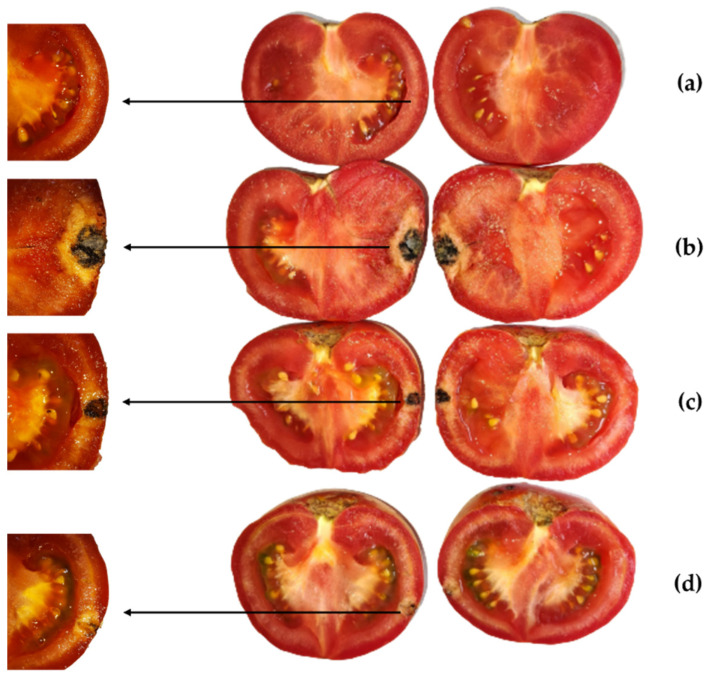
Internal lesions caused by *C. coccodes* on tomatoes cv. “Daniela” ten days after artificial inoculation in the presence/absence of *H. stoechas* inflorescence extract: (**a**) negative control; (**b**) fruits artificially inoculated with *C. coccodes* (positive control); (**c**) fruits treated with *H. stoechas* extract at 375 µg·mL^−1^; (**d**) fruits treated with *H. stoechas* extract at 750 µg·mL^−1^. Only one replicate per treatment is shown.

**Table 1 molecules-28-05861-t001:** Main absorption bands (expressed in cm^−1^) in the infrared spectrum of *H. stoechas* inflorescences.

Wavenumber (cm^−1^)	Assignment
3259	–O–H stretching (H-bonded)
2932	C–H stretching vibration
1688	C=O stretching
1652	C=O stretching/C=C stretching
1597	aromatic ring C=C vibration
1514	aromatic ring C=C vibration
1444	H–C–H asymmetrical bending
1367	symmetric methyl bending
1263	phenol –C–O vibration
1178	C–H in-plane bending/phenol –C–O vibration
1116	ring C–H bending
1069	C–O stretching vibration/C–O–C stretching vibration
984	–CH=CH_2_ groups vibration
925	CH_2_ rocking vibration
853	out-of-plane bending of =C–H bonds of an aromatic ring
812	C–H out-of-plane bending
781	C–H wagging mode
596	C–C in-plane bending

**Table 2 molecules-28-05861-t002:** Phytochemicals detected in *H. stoechas* inflorescence hydromethanolic extract, analyzed using gas chromatography–mass spectrometry (GC–MS).

RT (min)	Area (%)	Assignment	Qual
5.3068	1.7654	2-Cyclopenten-1-one, 2-hydroxy-	86
6.2920	2.6204	2-Hydroxy-*γ*-butyrolactone	32
6.4166	2.0342	1-Butoxypropan-2-yl isobutyl carbonate	43
7.6987	1.2980	1-Methyl-2,4,5-trioxoimidazolidine	43
7.8530	1.5041	1,3-Propanediamine, *N*-methyl-	56
8.0785	2.3624	Maltol	97
8.6127	2.0475	4*H*-Pyran-4-one, 2,3-dihydro-3,5-dihydroxy-6-methyl-	87
9.4793	3.5314	Catechol	97
9.6811	5.8441	Benzofuran, 2,3-dihydro-	87
9.7998	1.2578	5-Hydroxymethylfurfural	93
10.7019	2.4234	3(2*H*)-Pyridazinone, 6-methyl-	70
10.9987	1.9835	2-Methoxy-4-vinylphenol	91
12.7793	10.3925	4-Ethenyl-1,3-benzenediol	64
12.9930	2.2004	1-Acetyl-2-amino-3-cyano-7-isopropyl-4-methylazulene	53
13.5865	4.5524	*β*-d-Glucopyranose, 1,6-anhydro-	90
14.1325	1.4610	Dodecanoic acid	99
15.1712	5.6205	Quinic acid	87
15.3018	1.3040	d-Glycero-l-gluco-heptose	50
15.6995	2.1454	*α*-Bisabolol	64
17.9371	2.6290	4-Pyrimidinol, 6-(methoxymethyl)-2-(1-methylethyl)-	43
18.1448	1.4783	Hexadecanoic acid, methyl ester	98
18.5840	2.8538	Scopoletin	98
19.8364	1.5206	9-Octadecenoic acid (Z)-, methyl ester	99
20.1866	2.0290	Octadec-9-enoic acid	97
25.9082	5.0748	4*H*-1-Benzopyran-4-one, 3,5-dihydroxy-6,7,8-trimethoxy-2-phenyl-	94

RT = retention time, Qual = quality of resemblance.

**Table 3 molecules-28-05861-t003:** Effective concentration (EC) values (in µg·mL^−1^) against *A. alternata*, *C. coccodes*, *F. oxysporum* f. sp. *lycopersici*, *V. dahliae*, *R. solani*, and *S. sclerotiorum* obtained with the hydromethanolic extract of *H. stoechas* inflorescences.

EC	*A. alternata*	*C. coccodes*	*F. oxysporum* f. sp. *lycopersici*	*V. dahliae*	*R. solani*	*S. sclerotiorum*
EC_50_	279.3	177.0	185.1	182.6	75.7	87.0
EC_90_	481.3	276.6	372.8	330.4	106.9	132.0

**Table 4 molecules-28-05861-t004:** Suppression of mycelial growth using azoxystrobin, mancozeb, and fosetyl-Al (at manufacturer’s suggested dose and 1/10th of suggested one) for the examined fungal taxa.

Commercial Fungicide	Pathogen	Radial Growth of Mycelium (mm)		Inhibition (%)		Ref.
		Rd/10	Rd *	Rd/10	Rd *	
Azoxystrobin	*A. alternata*	49.4	38.9	34.1	48.1	This work
	*C. coccodes*	30.6	24.4	59.2	67.5	
	*F. oxysporum* f. sp. *lycopersici*	35.6	32.2	52.5	57.1	
	*R. solani*	50.6	17.2	32.5	77.1	
	*S. sclerotiorum*	14.0	9.0	81.3	88.0	[23]
	*V. dahliae*	26.0	24.0	65.3	68.0	[24]
Mancozeb	*A. alternata*	19.4	16.1	74.1	78.5	This work
	*C. coccodes*	0.0	0.0	100.0	100.0	
	*F. oxysporum* f. sp. *lycopersici*	0.0	0.0	100.0	100.0	
	*R. solani*	0.0	0.0	100.0	100.0	
	*S. sclerotiorum*	0.0	0.0	100.0	100.0	[23]
	*V. dahliae*	0.0	0.0	100.0	100.0	[24]
Fosetyl-Al	*A. alternata*	71.1	9.4	5.2	87.5	This work
	*C. coccodes*	0.0	0.0	100.0	100.0	
	*F. oxysporum* f. sp. *lycopersici*	67.8	4.4	9.6	94.1	
	*R. solani*	75.0	0.0	0.0	100.0	
	*S. sclerotiorum*	75.0	13.3	0.0	82.2	[23]
	*V. dahliae*	36.0	0.0	52.0	100.0	[24]

* In terms of recommended dose, Rd represents 62.5 mg·mL^−1^ of azoxystrobin (250 mg·mL^−1^ for Ortiva^®^, azoxystrobin 25%), 1.5 mg·mL^−1^ of mancozeb (2 mg·mL^−1^ for Vondozeb^®^, mancozeb 75%), and 2 mg·mL^−1^ of fosetyl-Al (2.5 mg·mL^−1^ for Fosbel^®^, fosetyl-Al 80%). The control (PDA only) exhibited a radial growth of the mycelium measuring 75 mm. All mycelial growth values provided are average values (*n* = 3).

**Table 5 molecules-28-05861-t005:** Lesion diameter (LD) and lesion size reduction (LSR) by *H. stoechas* inflorescence extract application on tomato fruits cv. “Daniela”, measured ten days after artificial inoculation with *C. coccodes*.

Treatment	LD (mm)	LSR (%)
Negative control	0	100
Positive control	42.2 ± 3.7	0
*H. stoechas* extract at 375 μg·mL^−1^	30.8 ± 3	27
*H. stoechas* extract at 750 μg·mL^−1^	7.8 ± 1.1	81.5

## Data Availability

The data presented in this study are available upon request from the corresponding author. The data are not publicly available due to their relevance to an ongoing Ph.D. thesis.

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
