# Peer review of "Helichrysum stoechas (L.) Moench Inflorescence Extract for Tomato Disease Management"

_molecules, 2023, doi:10.3390/molecules28155861_

Round 1

Reviewer 1 Report

In the manuscript entitled Helichrysum stoechas (L.) Moench Flower Extract for Tomato Diseases Management, the authors describe how they prepared one extract (methanol water) from the plant Helichrysum stoechas.
Why only one extract? In the case of extracts from multiple samples of the plant, it would be possible to compare the stability- the repeatability of the occurrence of individual components. Or, different methods of extract preparation could show differences.

The extract was analyzed using FTIR spectrometry and GC-MS and the major components were identified
This part seems to be performed and commented correctly

The antimicrobial activity of the extract was tested for 6 fungal species A. alternata, C. coccodes, F. oxysporum f. sp. lycopersici, R. solani, S. sclerotiorum, and V. dahliae
Standard methods were used and the results are interpreted correctly. What was the input for the probit analysis? Usually, at least 30 individuals in a group are needed for good results of this analysis; here there were only about 6. This is a significant limitation

The effect of fungicides on the growth of the fungal species used was tested.
I don't understand why this analysis was separated from the extract analysis and is interpreted in a different way. A direct comparison at the same concentrations could have better expressed the potential of the extract. Additionally, some values are just taken from other sources. It cannot be published in its actual form. The authors have to decide how to solve the problem of positive (fungicide) control.

A lesion formation test was carried out on tomato after inoculation with the fungus Colletotrichum coccodes and treatment with the extract in two doses, but no fungicide was used anymore. The chosen method of fruit inoculation is correct. The fruit treatment method allows precise quantification but is not practically applicable. It might have been better to use a surface application (spraying or dipping the fruit).

The purpose of the table in the appendix is perhaps to summarize the literature on plant extracts and their inhibitory effects on fungi and thus to support the uniqueness of the results obtained for the extracts. But this manuscript is not a literature review and therefore this table is redundant and should be omitted.
The entire manuscript is very long and the number of sources cited is unnecessarily high. The manuscript contains unnecessary formulations in the introduction and discussion sections that are not related to the problem at hand. It should be shortened before publication.
At least for bacteria, it is not appropriate to give the names of the describers after the Latin name.

Reviewer 2 Report

The manuscript describes the antifungal properties of methanol-aqueous extract from Helichrysum stoechas flowers (flowering heads) against six tomato pathogens. In addition,  infrared spectroscopy and GC-MS analyses of the extract were performed. The work brings some novelty to the eco-friendly strategy in horticulture.

Some issues arise:

l. 46. Essential oils should be omitted from the phenolic compounds

l. 52-57Please, rephrase the expression.

“Helichrysum Flowers” What do you mean? There are flowering heads. Accordingly, in the section “plant material” the used plant parts should be explain clearly.

l. 104 What is the reason to choose infrared spectroscopy for the analyses?

l. 119 Way the authors performed GC-MS of a methanol-aqueous extract? The extract is too polar for this chromatographic method. In my opinion the results are not reliable.

Provide the appropriate literature for both paragraphs 2.2 and 2.3.

l. 223 “flavonol” (not flavonone)

l.231 scopoletin is a simple coumarin

l.235 hydroxypyranone

l. 237 it is not correct to use “phenolic profile” – a few compounds are phenolics.

The authors should highlight the advantages of their approach in comparison with the essential oils application towards phytopatogens.

Extensive English editing is needed.

Round 2

Reviewer 2 Report

The Authors have addressed my comments from the first round. The manuscript has been improved according to all my suggestions. I have no more remarks.

Minor English editting

Author Response

The Authors have addressed my comments from the first round. The manuscript has been improved according to all my suggestions. I have no more remarks.

Response: We thank the Reviewer for his/her diligent evaluation and valuable feedback on our revised manuscript. We are delighted to hear that we have successfully addressed all his/her previous comments and improved the article according to his/her suggestions. His/her expertise and guidance have been instrumental in enhancing the quality of our research, and we sincerely appreciate his/her time and effort in reviewing our work.

Concerning the request for minor English editing, the manuscript has been thoroughly revised, and minor adjustments have been made.